# Tumor-Associated Macrophages/Microglia in Glioblastoma Oncolytic Virotherapy: A Double-Edged Sword

**DOI:** 10.3390/ijms23031808

**Published:** 2022-02-04

**Authors:** Sarah E. Blitz, Ari D. Kappel, Florian A. Gessler, Neil V. Klinger, Omar Arnaout, Yi Lu, Pier Paolo Peruzzi, Timothy R. Smith, Ennio A. Chiocca, Gregory K. Friedman, Joshua D. Bernstock

**Affiliations:** 1Harvard Medical School, Boston, MA 02115, USA; sarahblitz@hms.harvard.edu (S.E.B.); AKAPPEL@BWH.HARVARD.EDU (A.D.K.); nklinger@bwh.harvard.edu (N.V.K); oarnaout@bwh.harvard.edu (O.A.); ylu4@bwh.harvard.edu (Y.L.); pperuzzi@bwh.harvard.edu (P.P.P.); trsmith@bwh.harvard.edu (T.R.S.); eachiocca@bwh.harvard.edu (E.A.C.); 2Department of Neurosurgery, Brigham and Women’s Hospital, Boston, MA 02115, USA; 3Department of Neurosurgery, University Medicine Rostock, 18057 Rostock, Germany; Florian.Gessler@med.uni-rostock.de; 4Department of Pediatrics, University of Alabama at Birmingham, Birmingham, AL 35294, USA; gfriedman@uabmc.edu

**Keywords:** tumor-associated macrophages/microglia (TAMs), oncolytic virotherapy, tumor microenvironment, glioblastoma (GBM)

## Abstract

Oncolytic virotherapy is a rapidly progressing field that uses oncolytic viruses (OVs) to selectively infect malignant cells and cause an antitumor response through direct oncolysis and stimulation of the immune system. Despite demonstrated pre-clinical efficacy of OVs in many cancer types and some favorable clinical results in glioblastoma (GBM) trials, durable increases in overall survival have remained elusive. Recent evidence has emerged that tumor-associated macrophage/microglia (TAM) involvement is likely an important factor contributing to OV treatment failure. It is prudent to note that the relationship between TAMs and OV therapy failures is complex. Canonically activated TAMs (i.e., M1) drive an antitumor response while also inhibiting OV replication and spread. Meanwhile, M2 activated TAMs facilitate an immunosuppressive microenvironment thereby indirectly promoting tumor growth. In this focused review, we discuss the complicated interplay between TAMs and OV therapies in GBM. We review past studies that aimed to maximize effectiveness through immune system modulation—both immunostimulatory and immunosuppressant—and suggest future directions to maximize OV efficacy.

## 1. Introduction

### 1.1. Oncolytic Viruses

Oncolytic virotherapy centers on engineered viruses that target neoplastic cells. Oncolytic viruses (OVs) have shown great promise in the treatment of various cancer types due to their selective replication in cancer cells and the subsequent induction of tumor cell death [1,2]. This is possibly secondary to tumor cells’ impaired mechanisms of viral clearance [3,4,5,6] and/or through the genetic modification of OVs which enhance malignant cell selectivity [7]. Once in the tumor cells, OVs work through a mixed mechanism of induction of oncolysis and stimulation of antitumor immune activity [1,8]. Direct destruction of cells occurs, and lysis leads to release of viral particles, cytokines, and other cellular contents, and ultimately the induction of an immune response [1]. OVs can also be engineered to enhance direct tumor lysis through the delivery of suicide genes [9,10,11]. Furthermore, viral infection stimulates antitumor immune activity; after cell lysis, danger-associated molecular patterns (DAMPs), pathogen-associated molecular patterns (PAMPs), and cytokines are released that simulate the innate immune system [1]. Antigen presenting cells (APCs) recruit CD4+ and CD8+ T cells to create an adaptive immune response against infected tumor cells [12]. Immunogenic cell death (ICD) is a more recently discovered cell death modality that involves specific, timed changes to the cell surface (CRT expression) followed by release of soluble DAMPs (e.g., HMGB1, HSP). Importantly, this increases the immunogenicity of dying/dead cancer cells, and has the ability to potentiate adaptive immune responses and target residual cancer cells/tissues [13,14].

Almost all oncolytic viruses have some preference for infecting tumor cells over normal cells [15]. However, genetic engineering can be used to enhance OVs’ natural preference for cancer cells. For example, viruses can be re-targeted (genetically altered for tumor-specific viral entry) by preventing replication in non-dividing cells [7]. OVs can also be engineered to be armed (containing therapeutic transgene variants) to increase immunogenic reactions [16]. For example, arming HSV with interleukin-12 (IL-12) [17] or IL-4 [18] has demonstrated increased potency. 

For the treatment of glioma, three main viruses have been studied extensively: HSV, poliovirus, and adenovirus. HSV is the most broadly studied, with clinical trials in many cancer types including GBM [16]. HSV an attractive OV candidate for many reasons, including a large and highly stable genome, its potent cytolytic capability, high immunogenicity, and a convenient genome for genetic engineering; the availability of effective anti-herpetic drugs to treat adverse reactions is an added benefit [16,19]. Another common virus studied for glioma virotherapy is poliovirus. Therapeutic poliovirus (PVSRIPO) was created by combining a nonpathogenic poliovirus with a rhinovirus [20,21]. PVSRIPO demonstrated successful innate immune stimulation and cytotoxic cell recruitment in GBM [22,23]. Finally, adenoviruses have also been exploited for use in glioma. Adenovirus can function as a nonintegrating vector with relatively high capacity for gene delivery, including delivery of suicide genes [24,25,26,27]. It demonstrated successful induction of immune responses in GBM and caused direct oncolysis and autophagy [28]. 

Viral infection by any OV creates both an antiviral and antitumor response. In addition to direct oncolysis, OVs stimulate a host immune response. While tumor inflammation and immune cell recruitment is known to create an antitumor effect, it also prevents viral replication and distribution through its antiviral properties. The relative impact of these forces varies by cancer type [29]. Immune cell evasion is a classic characteristic of GBM [30], and OVs aim to reverse this by exposing the tumor to the innate and ultimately adaptive immune system.

### 1.2. Glioma-Associated Macrophages and Their Dichotomy

Tumor-associated macrophages/microglia (TAMs) are the most abundant non-neoplastic cell in the GBM tumor microenvironment (TME) and have been investigated as potential causes of OV therapy failure [31]. TAMs, which consist of both brain-resident microglia and bone marrow-derived myeloid cells from the periphery, constitute about 40% of the tumor mass in GBMs [32]. TAMs in the TME enhance tumor cell migration and invasion through secretion of chemotactic factors, enzymes, and cytokines [33,34]. As a result, there is a positive correlation between the number of TAMs and the malignancy of the brain tumor [35].

Classically, TAMs have been described in the M1/M2 dichotomy [36]. Although TAM phenotypes exist on a spectrum and are not truly dichotomous [37,38], this simplified classification provides a framework through which to understand their competing actions. Figure 1 shows a schematic representation of TAM polarization as well as the interaction with tumor growth and OVs. The classically activated M1 phenotype is activated by interferon gamma (IFN-γ) and lipopolysaccharides (LPS) through signal transducer and activator of transcription 1 (STAT1) activity [12,39]. M1 TAMs promote strong IL-12 mediated T helper 1 (Th1) responses and activate natural killer (NK) cells through pro-inflammatory cytokine production, including tumor-necrosis factor alpha (TNF**-**α), IL-β, IL-6, IL-8, IL-12, and IL-23, regulated in part by the nuclear factor kappa B (NF-κB) pathway [12,37,40,41,42]. They are also capable of phagocytosis and antigen processing and presentation, making them a bridge between innate and adaptive immune systems [12,39,43]. This creates a systemic and durable immunity to tumor cells [8]. 

M2 phenotype TAMs are alternatively activated by peroxisome proliferator-activated receptor-γ (PPARγ) and STAT6, which suppress the NF-κB pathway [39]. They promote strong Th2-associated effector functions and induce regulatory T cells (Tregs). Through the production of IL-1RA, IL-10, vascular endothelial growth factor (VEGF), and transforming growth factor beta (TGF-β), M2s stimulate tissue remodeling and tumor development [37,39,44,45,46,47,48]. They also lead to the resolution of inflammation through high endocytic clearance capacities and trophic factor synthesis, angiogenesis, and tumor proliferation and progression [12,33]. Overall, the phenotypes are primarily regulated through either interferon regulatory factor 4 (IRF4) or IRF5, competing through Toll-like receptor (TLR) signaling, and polarizing cells towards the M2 or M1 phenotype, respectively [49,50]. This demonstrates that the in vivo TAM phenotype is based on the dominant cytokines, trophic factors, and proteins in the microenvironment.

### 1.3. Glioma-Associated Macrophages and Immunosuppression

In GBM, there is general immunosuppression in the TME. Glioma cells and TAMs work symbiotically—glioma cells attract TAM infiltration and TAMs promote glioma growth and invasiveness [41,51,52]. TAMs are recruited to the TME by chemoattractants from glioma cells, such as CC chemokine ligand 2 (CCL2)[53,54] and soluble colony-stimulating factor 1 (sCSF-1) [55,56]. Polarization to M2 occurs through a variety of cytokines, including IL-10, IL-4, IL-6, macrophage colony stimulating factor (M-CSF), TGF-β, and prostaglandin E2, and the TAMs become immunosuppressive [37,51,52,57,58,59]. Interactions between TAMs and other immune cells in the TME provides additional mechanisms of immunosuppression. TAMs produce chemokines that recruit Tregs, and they both secrete IL-10, which interferes with IFN-γ production and impairs infiltrating T cells [60,61]. TAMs also upregulate several surface molecules that inhibit T cell activation and induce T cell apoptosis including cluster of differentiation 95 (CD95), CD70, and programmed cell death ligand 1 (PD-L1) [30,62]. This leads to fewer tumor-infiltrating immune effector cells (Teffector) and prevents an immune attack against the glioma [30,62].

## 2. Discussion

### 2.1. Polarizing TAMs towards the M1 Phenotype

Polarizing TAMs towards the M1 phenotype and creating a pro-inflammatory TME is a key mechanism of the OV-induced antitumor effect. This polarization and increased inflammatory response are seen in a variety of viruses in GBM, including HSV [63,64,65,66], adenoviruses [67,68,69], parvoviruses [70,71,72], and vaccinia viruses [73]. This coincides with an increased Teffector to Treg ratio, likely related to an IFN-γ influx [65]. Previous studies have utilized this mechanism in altering viruses to enhance the M1 phenotypic shift. IL-12 is the most commonly used cytokine to enhance antitumor efficacy of OVs [74,75]. As seen in Figure 1, IL-12 plays a role in the antitumor response, through induction of Th1 differentiation, stimulation of NK growth and cytotoxicity, IFN-γ release, and angiogenesis inhibition [75,76]. Arming HSV with IL-12 has demonstrated increase in IFN-γ and reduction in Tregs in tumors, along with increased survival in murine models [17,76,77,78,79]. Other cytokines have also been used, including FMS-like tyrosine kinase 3 ligand (Flt3L), which is associated with increases in intratumoral dendritic cells and CD8+ T cells [80,81] and improves survival in glioma-bearing mice [82]. The addition of IL-4 to HSV also increased infiltration of macrophages and CD4+ and CD8+ T cells into murine models of intracranial glioma and prolonged survival [18]. In contrast however, transfection of HSV expressing IL-10, which is anti-inflammatory, did not produce a significant survival advantage compared to saline-treated controls in the same animal model and did not result in increased infiltration of CD8+ T cells [18].

### 2.2. Combination Therapies with Checkpoint Inhibitors

Combination therapies have also been explored to enhance immunostimulation with OVs. Checkpoint inhibitors, such as anti-CTLA-4 (cytotoxic T-lymphocyte-associated protein 4) and anti-PD-1, have demonstrated efficacy in many cancer types. They enhance activation of T helper cells and effector cells while suppressing Tregs [83]. Anti-CTLA-4 and anti-PD-1 antibodies work through distinct and non-redundant inhibitory pathways on immune cells [65,83]. Unfortunately, they have not resulted in significant benefits in GBM clinical trials [84,85]. This is related in part to the immunologically “cold” and highly immunosuppressive TME [86]. However, triple combination of HSV armed with IL-12 along with anti-CTLA-4 and anti-PD-L1 increased influx of macrophages and M1 polarization and was very effective in curing both murine glioma models including aggressive carcinogen-induced tumors [65,87]. Immune checkpoint modulation has also been used in adenovirus armed with the immune costimulatory OX40 ligand, which activates lymphocytes and leads to proliferation of CD8+ T cells [88]. Adding anti-PD-L1 antibody with this OV also demonstrated significantly increased survival in mice with gliomas [88]. Adenovirus armed with costimulatory ligand glucocorticoid-induced tumor necrosis factor receptor (TNFR) family-related ligand (GITRL) also increased CD8+ T cell infiltration and increased survival in mice with gliomas [89]. Clearly, the inflammatory role of M1 TAMs, which go hand in hand with the various mechanisms of armed viruses, is crucial to effectiveness of OVs. 

### 2.3. Combination Therapies with Immunosuppressive Medications

However, TAMs and M1 polarization also play a detrimental role in OV efficacy through their antiviral activity, which is not virus-specific [31]. Gliomas treated with HSV have shown intratumoral clearance of over 80% of viral particles shortly after delivery, which is associated with up-regulation and infiltration of TAMs [90,91]. TAMs can directly uptake viruses through endocytosis or reduce replication through secretion of antiviral cytokines [64,66,92]. Upon injection, there is an immediate M1 antiviral reaction competing with glioma cells for virus uptake [93,94]. Within 72 h of HSV injection, there is a significant decrease in the volume of tumor cells that contains HSV-mediated gene expression [91]. However, depletion of the innate immune response including peripheral phagocytic cells and brain microglia via immunosuppressants can enhance intratumoral uptake and spread of OV [31,91]. TAMs are also able to create a non-permissive barrier that prevents OV replication and dissemination, which leads to viral gene silencing [64]. This detrimental TAM-induced antiviral effect has been supported in studies that demonstrated that blocking STAT1/3 activity rescues OV replication and enhances the therapeutic efficacy of oncolytic virotherapy [64]. In addition, chemical ablation of TAMs in glioma-bearing rodent models enhanced the antitumoral effects of HSV [31]. Decreasing TAMs, despite their antitumoral proinflammatory effects, can increase OV uptake, replication, and efficacy.

Due to the known deleterious effects of TAMs related to early viral clearance, many studies have aimed to suppress recruitment of TAMs with OVs. For example, cyclophosphamide inhibits the production of IFN-γ by natural killer (NK) cells and reduces the concentration of TAMs [91]. Even residual TAMs demonstrated suppressed expression of antiviral cytokines [95,96]. This led to a 10-fold increase in intratumoral OV gene expression [91]. Preadministration of cyclophosphamide in athymic mouse models of human glioma, caused increased OV uptake and intratumor distribution allowing for reduced OV doses, reduced tumor burden, and increased survival [97]. Other ways to decrease IFN signaling and pro-inflammatory cytokine induction in tumor cells, therefore limiting TAM recruitment, includes administration of rapamycin to block integrin beta 1 receptors [98], which are expressed on the cell surfaces of macrophages [99]; pretreatment with the histone deacetylase inhibitor valproic acid [100]; or administration of cellular communication network factor 1 (CCN1) antibodies [101,102,103]. The addition of TGF-β can also suppress TAM recruitment by dampening the innate immune response through cell growth inhibition and apoptosis via transcriptional induction of genes, such as cyclin-dependent kinase inhibitors (CDKIs) [104,105]. This inhibits NK and TAM recruitment, activation, and function, thereby enhancing OV replication [105].

Other techniques have focused more on inhibiting TAMs that are present in the TME. Clodronate encapsulated in liposomes is taken up by phagocytic cells and results in intracellular accumulation and TAM apoptosis, thereby depleting the TAM population [106,107,108]. In murine GBM, this led to a five-fold increase in viral replication [31]. M1 TAM mechanisms can also be restricted, such as blocking brain angiogenesis inhibitor [63,109] or STAT1/3 activity [43,64,110]. In addition, depleting NK cells, which coordinate TAM activation in response to OV, has a beneficial effect [105,111]. 

Finally, some studies have investigated ways to bypass the antagonizing effects TAMs have on virotherapy regardless of their presence. OVs can be transported with carrier cells, protecting them from neutralization and opsonization and assisting with homing to the tumor site in studies with systemic injection [112,113,114,115,116]. Overall, finding ways preclinically to restrict TAM function, whether through reduced recruitment, reduced activation and function, or preventing interactions with OVs demonstrates that inhibiting TAMs has potential to benefit OV efficacy.

## 3. Future Directions

One of the key next steps is to investigate the combined mechanisms that prevent initial antiviral TAM actions, while still allowing for later TAM-directed antitumor responses. The balance between these two opposing functions in part determines the efficacy of OV therapy. However, modulating these responses, which often encompass overlapping immunological pathways, is challenging and poorly understood [8,117]. It is possible that many of the previously mentioned tactics can be combined to create both antiviral and antitumor responses. For example, selective and transient immunomodulation with immune-inhibiting therapeutics may still allow for sufficient tumoral infection to induce a robust antitumor response.

## 4. Conclusions

Understanding the mechanisms that inhibit and potentiate oncolytic virotherapy is necessary for this promising therapy to reach its full potential. Overall, TAMs function in OV therapy as a double-edged sword. They play a crucial role in the immune stimulation that creates the antitumor response generated by OVs. Initial innate immune responses orchestrate subsequent lasting adaptive immune responses. However, TAMs also inhibit efficient intratumoral viral distribution. Both immunostimulatory and immunosuppressant adjuvants have shown benefits in OV research. More research on combination therapies is necessary to find cooperative tactics. Harnessing TAMs to promote both antiviral and antitumor effects will optimize OV efficacy in the future.

## Figures and Tables

**Figure 1 ijms-23-01808-f001:**
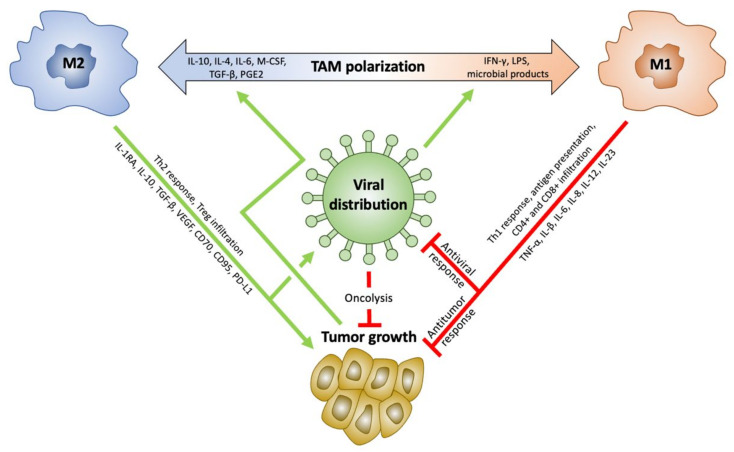
Schematic representation of the interaction between M1/M2 tumor-associated macrophage/microglia (TAM) polarization with oncolytic viral distribution and tumor growth in glioblastoma multiforme. Elements of the microenvironment influence polarization between TAM subsets of immuno-suppressive M2 and immune-stimulatory M1. Tumors polarize TAMs towards the M2 phenotype, which support tumor growth. Oncolytic virotherapy (OV) inhibits tumor growth through two main mechanisms: direct oncolysis and the antitumor response. The latter is a result of virus-induced polarization towards the M1 phenotype, which creates an immune response against tumor growth. However, this also stimulates an antiviral response, limiting the beneficial effects of OVs. IL interleukin, M-CSF macrophage colony stimulating factor, TGF-β transforming growth factor beta, PGE2 prostaglandin E2, IFN-γ interferon gamma, LPS lipopolysaccharide, Th T helper, Treg T regulatory, VEGF vascular endothelial growth factor, CD cluster of differentiation, PD-L1 programmed cell death ligand 1, and TNF-α tumor necrosis factor alpha.

## Data Availability

Not applicable.

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
