# Peer review of "Tumor-Associated Macrophages/Microglia in Glioblastoma Oncolytic Virotherapy: A Double-Edged Sword"

_ijms, 2022, doi:10.3390/ijms23031808_

Round 1
Reviewer 1 Report
Blitz et al. deliver a concise and yet comprehensive review of the potentials and pitfalls of developing oncolytic virotherapy to treat GBM.
The work clearly summarizes the mechanistic aspects of this approach acknowledging that a better understanding of the complex immunomodulatory landscape is necessary for this promising therapy to reach its full potential
The manuscript certainly deserves to be published after considering the comments below.
Recent and less recent papers have linked the efficacy of oncolytic virus-based immunotherapy of cancer to the induction of immunogenic cell death (ICD), a relatively newly recognized regulated cell death modality (e.g., PMID: 31969562). It would be appropriate to mention such a notion in the introduction of the paper.
Line 205 – typo, please replace “Fro” with “For”.
Author Response
Dear Reviewers and Editors,
Thank you for taking the time to review our manuscript entitled “Tumor-associated macrophages/microglia in glioblastoma oncolytic virotherapy: A double-edged sword.” We truly appreciate the expert review and positive feedback.
For the revisions, we focused mainly on organization and introducing structure to facilitate following the topics with ease. We added the recommended subheadings to the introduction and discussion and split up paragraphs with too many ideas. We better organized some paragraphs that did not have a logical flow.
In addition, as recommended, we mentioned immunogenic cell death in the introduction in its relation to OV and have added the pertinent reference. We also clarified that all OVs have some preference for tumor cells, and all can be improved with genetic engineering. We provided examples for how to increase preference for glioma cells and arming viruses to make them more potent, which is expanded upon later in the text.
Again, we appreciated your comments and suggestions and look forward to our paper being made available to our colleagues in Neuro-Oncology.
Respectfully,
The Authors
Corresponding and Reprint Author:
--
Joshua D. Bernstock, MD, PhD
Department of Neurosurgery
Brigham and Women’s Hospital
jbernstock@bwh.harvard.edu
Reviewer 2 Report
The review focuses on the important and timely topic of the dual role that glioma-associated macrophages play in oncolytic virotherapy outcomes. The text is dense, but logical and very informative. The graphic illustration that accompanies the text is excellent. The list of references includes the most important studies to date.
Minor revision suggestions
Introduction
The presence of subheadings can greatly facilitate the perception of the text for readers. The structure and main ideas will become immediately visible. There are three logical subsections in the “Introduction” section and I suggest adding titles to these subsections. The following titles are possible: 1) Oncolytic virotherapy (lines 28-67), 2) Glioma-associated macrophages and their dichotomy (lines 68-100). 3) Glioma-associated macrophages and immunosuppression (101-114).
Lines 43-46. That doesn't seem to me to be the right dichotomy. Almost all oncolytic viruses have some preference for tumor cells and almost all oncolytic viruses can be improved by genetic engineering. I would say that the variable degree of natural OV preference for cancer cells could be improved by genetic engineering, which could also provide additional means for therapeutic potency. After this phrase, I would give examples of how it can be done, therefore some examples can be added to the relevant phrase (lines 47-49).
Lines 50-60. It would be useful to structure the corresponding fragment of text logically, because the three viruses are described in this fragment in a very chaotic way.
More suggestions for the text improvement are in the attached pdf file.
Discussion
For better understanding and easier presentation of the information in this section, I suggest titling the subsections. There are three logical subsections in the “Discussion” section. A potential structure with subsections headings could be as follows: 1) Polarizing TAMs towards the M1 phenotype (lines 116-134), 2) Combination therapies with check point inhibitors (lines 135-153), 3) Combination therapies with immunosuppressive medications (lines 154-197).
The text fragment (lines 172-197) contains too many ideas crammed into one paragraph. It would benefit from being divided into several paragraphs using the "one idea, one paragraph" principle.
More suggestions for the section improvement are in the attached pdf file.
Author Response

(The authors gave the same response as above.)
